# Integration of Passivated Gold Mirrors into Microfabricated Alkali Vapor Cells

**Florian Wittkämper ***, **Theo Scholtes**, **Sven Linzen**, **Mario Ziegler** and **Ronny Stolz**

Leibniz Institute of Photonic Technology, Albert-Einstein-Straße 9, D-07745 Jena, Germany;
theo.scholtes@leibniz-ipht.de (T.S.); sven.linzen@leibniz-ipht.de (S.L.); mario.ziegler@leibniz-ipht.de (M.Z.);
ronny.stolz@leibniz-ipht.de (R.S.)
* Correspondence: florian.wittkaemper@leibniz-ipht.de

**Abstract:** Measurements of weak magnetic fields demand a small distance between the sensor and the to-be-measured object. Optically pumped magnetometers (OPMs) utilize laser light and the Zeeman effect in alkali vapor cells to measure those fields. OPMs can be used in transmission or reflection geometry. A minimization of the distance between active volume and magnetized source calls for reflection geometry with integrated mirrors. Unfortunately, cesium reacts chemically with most materials, especially high-performing materials, such as gold. Herein, we show the first functional OPM cell using a gold mirror inside the cell. We fabricated the gold mirrors with and without a passivation layer in order to evaluate the feasibility of expanding on the limited list of possible mirror materials. A comparison of this implementation revealed that mirrors without a passivation layer only reach a reflectivity of about 6% while mirrors with a passivation layer retain reflectivity values of about 90% in the visible light to near-infrared spectrum. This result and the proof of elemental cesium in the alkali vapor cell demonstrates the feasibility of passivated gold mirrors for applications in alkali vapor cells for OPMs.

**Keywords:** alkali vapor cell; coatings; atomic layer deposition; mirrors; optically pumped magnetometer





## 1. Introduction

Optically pumped magnetometers apply the Zeeman effect to measure magnetic fields at high resolution. To utilize the Zeeman effect, they are pumped with a laser beam of adequate wavelength [1]. The heart piece of an OPM setup is the alkali vapor cell. This cell is usually heated to increase the amount of evaporated alkali metal. Due to the high sensitivity and absence of cryogenic cooling, OPMs are well suited for biological applications.

The alkali vapor cells are typically implemented using two different technologies: the first fabrication method is glass blowing; the second, more flexible method utilizes technology from the area of micro-electromechanical systems (MEMS). Such MEMS cells allow for higher reproducibility and miniaturized system integration because of their basic thin-film fabrication technology in comparison to glass-blown cells [2,3].

In order to measure weak magnetic field amplitudes with high spatial resolution, the distance between the sensor and magnetized source should be minimized. Although OPMs can be used in general in transmission or reflection setup, in both cases optical components must be placed between the sensor and source. These components increase the distance in between the MEMS cell and magnetized source in the case of transmission due to the photodiode and in case of reflection due to the mirror. Therefore, an integration of the mirror into the cell would significantly reduce the distance between the sensor and magnetized source for the reflection geometry. Furthermore, to reduce the number of optical transitions, the mirror should be implemented into the alkali vapor volume of the MEMS cell (refer to Figure 1).

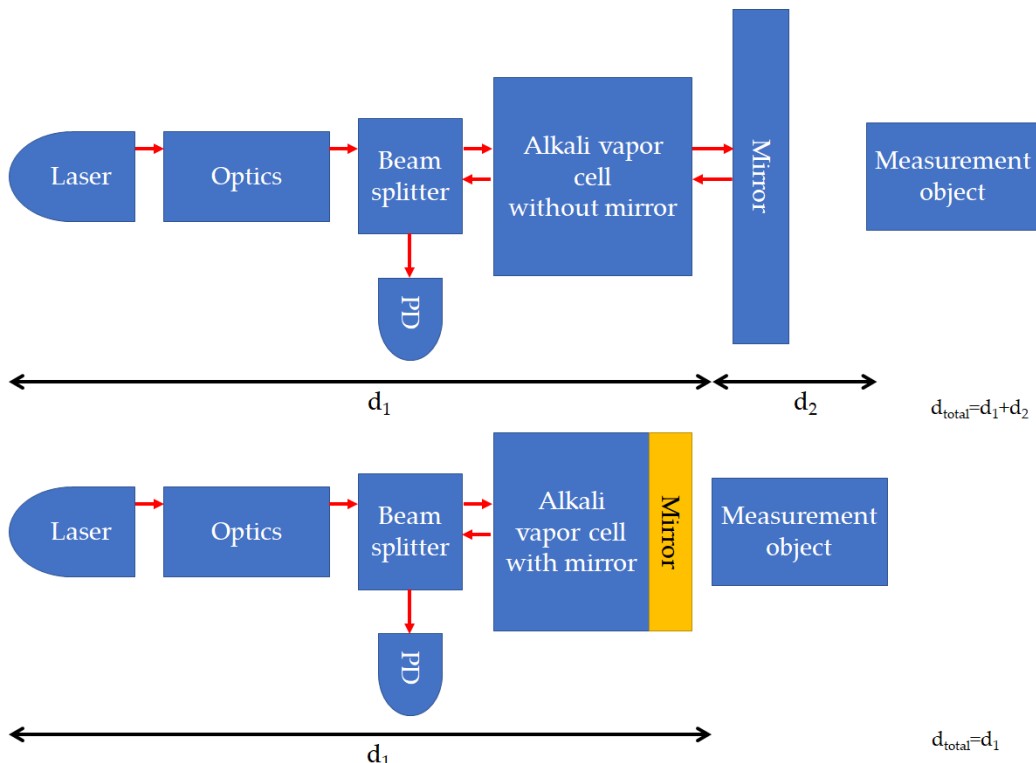

**Figure 1.** Schematic setup with an external mirror (**top**) and with an in-cell mirror (**bottom**). The distance between the measurement object and alkali vapor cell can be reduced significantly using the mirror inside the cell. PD: photodiode.

The in-cell mirrors can be realized in various ways, such as alkali-resistant metallic [4–6] or dielectric mirrors, Bragg gratings [7,8], or non-alkali-resistant but passivated mirrors [9]. While dielectric mirrors induce stress on the glass wafer on which it is deposited, alkali-resistant metals, such as aluminum, have a low reflection [10]. While gold has a high reflection coefficient (larger than 90%) within the wavelength range of interest (700–900 nm) it is not resistant to alkali metals such as cesium or rubidium [11]. Unfortunately, gold reacts with cesium to form cesium auride, which leads to a strong change in the optical performance [12,13]. A suitable passivation might therefore be vital for the function of the mirror and needs to be developed. Simultaneously, the degree of color change of the gold mirror of the cell fabricated in such a way serves as an indicator for the quality of the passivation and the applicability of the mirror and cell. This implies that the color of mirrors with a passivation of low quality or without passivation will therefore exhibit a change in a larger area than the color of mirrors with a high-quality passivation.

## 2. Materials and Methods

Standard boron-doped silicon wafers (MicroChemicals, Ulm, Germany). with a diameter of 4 inches and a thickness of 525 µm were etched using potassium hydroxide (KOH) to fabricate cavities of $8 \times 8$ mm$^2$ for the alkali atom cells. These cavities were later encapsulated by two bond processes with Borofloat-wafers (Wafer universe, Elsoff, Germany) glass windows, as described in [14]. Thus, glass wafers with a diameter of 4 inches and a thickness of 700 µm were lithographically processed to realize $4 \times 4$ mm$^2$ squares for the gold mirrors. The 100 nm thick gold mirrors and an adherent layer of Ti were deposited using thermal evaporation. The individual mirrors were realized within the whole wafer area using a lift-off resist mask. After lift-off, the glass wafer was annealed at 400 °C for 36 h in air. Subsequently, a 10 nm thick aluminum oxide passivation layer was deposited via atomic layer deposition (ALD) [15,16]. This glass wafer with optical mirrors

was afterwards anodically bonded to the silicon wafer at below $4 \cdot 10^{-6}$ mbar at 350 °C. This implementation was developed based on the process description from [14].

The mirrors also need to withstand the bonding processes required for the closure of the cavities. In Figure 2, the parameters of the first anodic bond are shown (the second anodic bond is not shown, but is implemented using the same parameters). The time scale was normalized so that the time at 0 s precedes the start of the anodic bond by 5 s. To reduce voids in the bond interface, the anodic bond was performed in high vacuum using distance flags between the two bond specimens to evacuate the bond interface. The anodic bond was performed in three subsequent steps. This led to bond current amplitudes well below the maximum of the current source (50 mA at 600 V), allowing for an analysis of the bond current's behavior. The three peaks in the bond current (current in Figure 2) show a fast amplitude rise followed by an exponential decay, as is expected for an unhindered bond [17]. The temperature was chosen to ensure a fast anodic bond process.

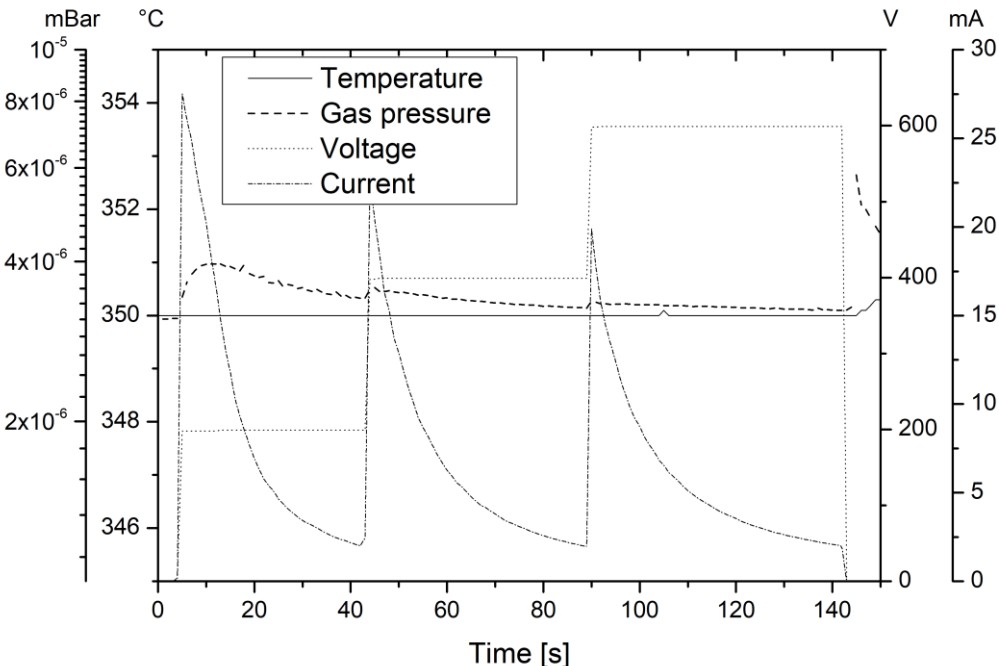

**Figure 2.** Time dependencies of the first anodic bonding process parameters.

After the first anodic bond, a cesium azide solution was pipetted into each cell cavity. To hermetically seal the alkali vapor cells, a second unprocessed glass wafer was anodically bonded from the opposite silicon wafer side using the same process parameters as employed for the first anodic bond. After dicing the completed wafer package into separate alkali vapor cells, microscopic images of their mirrors were taken. Then, the cesium azide within the individual cells was decomposed using photolysis by means of a pulsed excimer laser beam with a wavelength of 248 nm. An aperture was used to protect the passivated gold mirrors from the excimer laser beam. Afterwards, those cells were annealed at 80 °C for at least 24 h to increase the speed of the reaction of cesium and gold. Finally, the reflectivity of the gold mirrors was measured, and microscopic images were taken.

For the anodic bonding process, we used an EVG®501 Wafer bonding system [18]. The optical measurements were undertaken using a confocal spectrometer Olympus USPM [19].

### 3. Results

Two wafers, each with 16 mirror cells, were fabricated. After the decomposition and annealing step, the cells without a passivation layer showed a significant change in color, indicating a chemical reaction between gold and cesium. The cells with a passivation layer showed minimal change in color compared with a cell without decomposed cesium

azide (see Figure 3). The light grey areas in the bond interface (refer to the outer frames in Figure 3) are the aforementioned voids. No particles could be observed at the center of those voids, indicated by interference fringes, by using a light microscope. Therefore, their main source is trapped gas. The dark substance near the optical mirror in the middle panel of Figure 3 is a mixture of different secondary products of the photolysis as well as gases reacting with the emerging cesium.

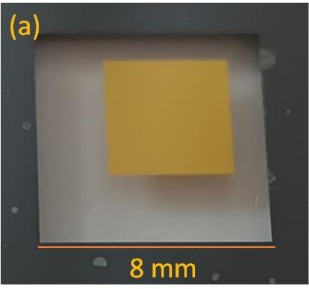
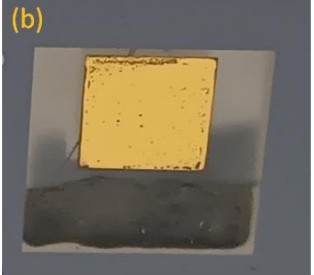
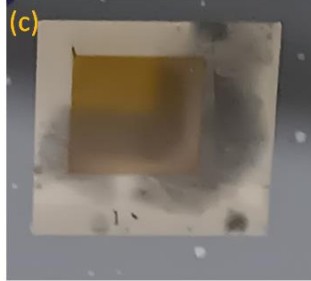

**Figure 3.** Photographs of fabricated cells. Cell (**a**) is the refence cell without decomposition, serving as a comparison for the other two. Cells (**b**,**c**) are shown after the steps of decomposition and annealing. Cell (**b**) has a passivation layer, while cell (**c**) does not. See the main text for more details.

The measured reflectivity of uncoated gold mirrors (Figure 4, panel (a)) shows a drop in reflectivity in the whole visible spectrum to values below 6%. Simultaneously, the passivated gold mirror retains reflectivity values above 90%. The reflectivity values for the wavelength of the cesium D1 line, which is important for the implementation of OPMs in this work, is 3.7% for unpassivated gold mirrors and 91% for passivated gold mirrors (refer to Figure 4). The measurements were taken from the back of the cells (i.e., through the backside of the mirror's glass wafer) in order to exclude any effect from cesium, cesium azide, or other cesium compounds inside the cells.

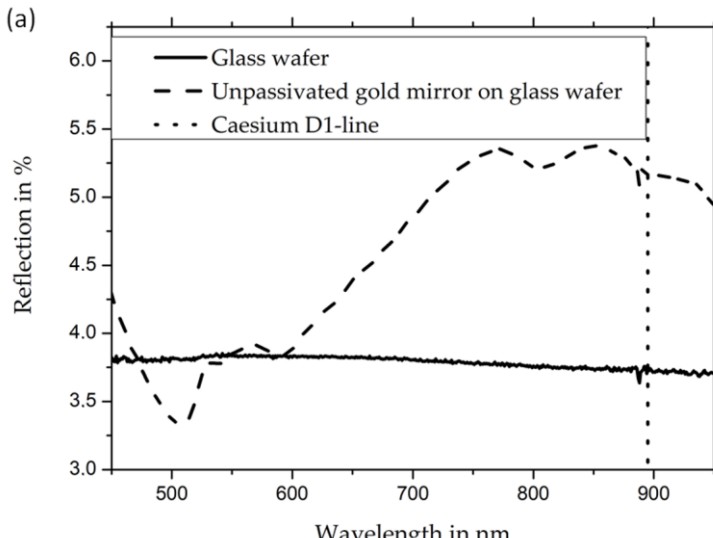

**Figure 4.** *Cont*.

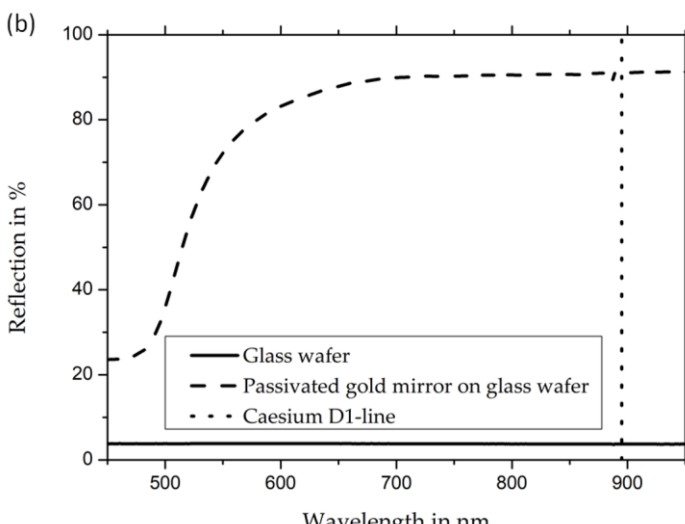

**Figure 4.** Measurement of reflectivity after the decomposition of cesium azide and the annealing of a mirror without (**a**) and with (**b**) passivation.

To validate the quality of the gold mirror, microscopic images of uncoated and coated gold mirrors before and after the anodic bonding, decomposition, and annealing at 80 °C for 24 h were made (see Figure 5). The images were taken from the back of the cell. The grey structures beside the mirrors are cesium azide. While the unpassivated gold mirror reacts nearly completely with cesium to form cesium auride (Figure 5a,b), the coated mirror only shows localized dark spots (Figure 5, panel (d)). This implies that the passivation coating is damaged by the anodic bonding process. At the same time, these defects affect only a small portion of the mirror. Thus, most of the mirror area, and therefore its function, remains intact.

The most likely explanation for the formation of holes in the coating is the thermally induced mechanical stress within the thin film system caused by the huge differences of the thermal expansion coefficients between the glass substrate ($3.3 \cdot 10^{-6}$ K$^{-1}$), the gold layer ($14.2 \cdot 10^{-6}$ K$^{-1}$), and the alumina layer ($6.5\text{–}8.9 \cdot 10^{-6}$ K$^{-1}$) [10,20,21].

To evaluate the cell concerning the basic function and buffer-gas-dependent application of the cell, a transmission spectrum was measured. The setup for this measurement is schematically depicted in Figure 6 (on the right side). The transmission spectrum shows the absorption peaks of elemental cesium of the alkali vapor cell with a passivated mirror as well as a reference cell with four distinct absorption peaks. The existence of the absorption peaks proves the presence of elemental cesium. This is a requirement for the performance of an alkali vapor cell for OPM setups and for proving the feasibility of this cell for OPMs. Furthermore, the measured spectra are fitted with four peaks, each displaying a Voigt profile. By comparing the nitrogen-induced shift of those peaks of the mirror cell compared to the reference cell, the buffer-gas pressure can be calculated. Thus, the value of the buffer-gas pressure was derived (see Figure 6). For the shown measurement (Figure 6, left), the buffer-gas pressure is about 372 mbar. As a consequence, the buffer-gas pressure leads to a merge of cesium absorption peaks, allowing for the pumping of several optical transitions at once, which is necessary for measurement principles such as the light-shift-dispersed Mz mode [22].

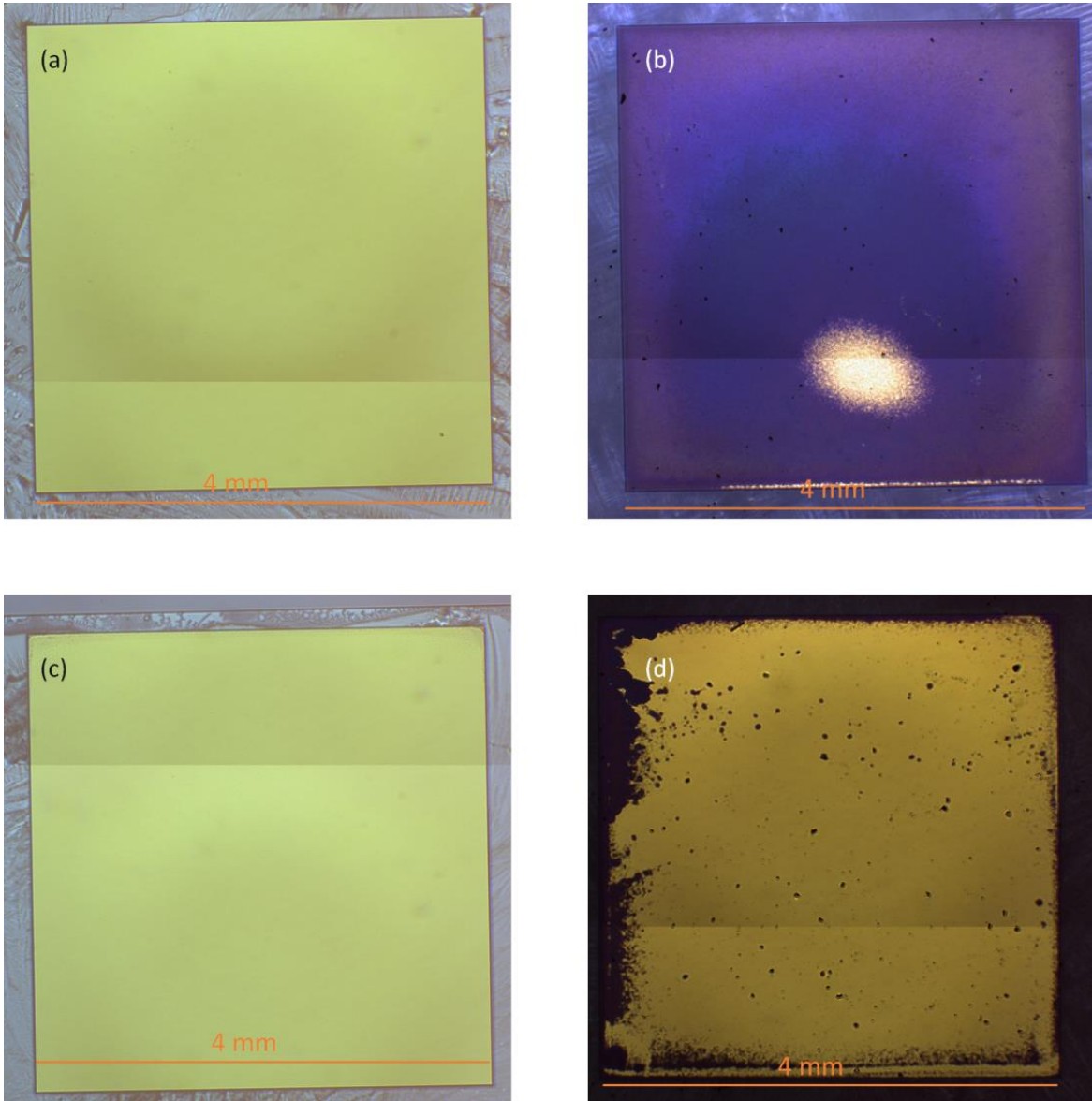

**Figure 5.** Microscopic images of mirrors before (left column) and after (right column) decomposition of cesium azide and annealing with (bottom row) and without (top row) passivation. Panel (**a**) depicts a gold mirror before photolysis, where no chemical reaction between cesium and gold can have occurred. The gold mirror in panel (**a**) does not have a passivation and is indistinguishable from the mirror in panel (**c**) which has a passivation and where the step of photolysis is yet to be done. In panel (**b**) an unpassivated mirror after photolysis and annealing at 80 °C is shown, wide areas of the gold have reacted with the cesium. In panel (**d**) a passivated gold mirror after photolysis and annealing can be seen. This mirror is mostly intact, but several small areas of chemical reaction between cesium and gold are visible.

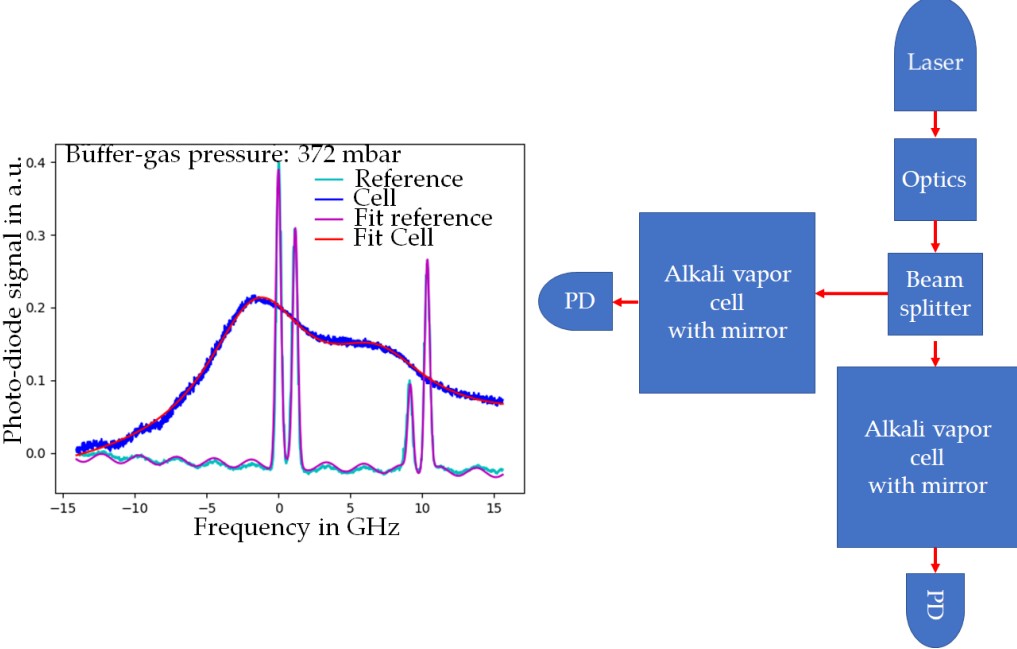

**Figure 6.** Left: Transmission spectrum of the passivated mirror cell for the evaluation of the buffer-gas pressure. Blue: measurement of the mirror cell; turquoise: measurement of a reference cell without mirror; red: calculated fit of the mirror cell; purple: calculated fit of the reference cell. The shift in the spectrum is calculated to determine the buffer-gas pressure. Right: Schematic setup of the reference measurement for the calculation of the buffer-gas pressure. PD: photodiode.

## 4. Discussion

Functional MEMS alkali vapor cells with gold mirrors inside the optical window have been fabricated for the first time. They enable miniaturized sensor heads for new OPMs. To evaluate the effect of an aluminum oxide coating on the performance of the gold mirror in a cesium atmosphere, the parameters of coated and uncoated cells were compared. The implementation of a passivation coating led to visible effects even in photographs. The uncoated gold mirror changed its color, indicating a negative effect on the reflectivity. Optical measurements showed that the uncoated mirrors reflected less than 6% of the irradiated light, while the coated mirrors retained a reflectivity of above 90% in the visible to near infrared spectrum. At 895 nm, the D1 line of cesium, the reflectivity of the passivated mirror retained a value of 91%, while the unpassivated mirror only reflected 5% of incoming light with a wavelength of 895 nm. The measured cesium spectrum demonstrated that these cells are well suited as miniaturized alkali vapor cells for the intended OPMs. It shows the peaks, which are characteristic for remaining elemental cesium in the cell. Thus, the passivation increases the lifetime of the gold mirror drastically even within ambient cesium vapor.

Even though all experiments were done with cesium as the alkali metal, it stands to reason that this passivation layer is applicable for other alkali metals like e.g., rubidium as well. $Al_2O_3$ deposited via ALD is used in both [15,16] to reduce the rate of reaction between alkali metal and the surrounding glass. Therefore, the passivation effect on gold mirrors can be expected to be the same.

## 5. Conclusions and Outlook

The passivation process was established successfully to protect gold mirrors in a cesium atmosphere for implementation purposes. Even though a reflection geometry for OPMs can be realized with the presented cells, a further improvement of the passivation is necessary.

To further reduce even localized defects, an iterative annealing and coating process is proposed. This process could consist of the annealing of gold (step 1), the deposition of

alumina (step 2), an annealing step of the covered gold (step 3), and an optical inspection to verify the absence of defects in the passivation coating (step 4). In the event that step 4 was not successful, steps 2–4 can be repeated until the passivation coating is free of defects.

**Author Contributions:** Conceptualization: F.W. and T.S.; methodology: F.W. and T.S.; investigation: F.W.; resources: R.S.; writing—original draft preparation: F.W.; writing—review and editing: T.S., S.L., M.Z. and R.S.; visualization: F.W.; supervision: R.S.; project administration: R.S.; funding acquisition: R.S. All authors have read and agreed to the published version of the manuscript.

**Funding:** We gratefully acknowledge the partial financial support from the Federal Ministry of Education and Research (BMBF) of Germany under Grant No. 033R130EN (DESMEX II) and 13N15436 (OPTEM). The technological development was partly also supported by the Free State of Thuringia under the number 2021 FE 9087 and co-financed by European Union funds within the framework of the European Regional Development Fund (ERDF) and REACT-EU.

**Institutional Review Board Statement:** Not applicable.

**Informed Consent Statement:** Not applicable.

**Data Availability Statement:** Not applicable.

**Acknowledgments:** We thank Birger Steinbach for the evaporation of the gold mirrors, Katrin Pippardt for the lithography and wet etching of silicon, and Valentin Ripka and Hanjörg Wagner for the deposition of alumina via ALD. We thank Tim Kügler for writing the code to calculate buffer-gas pressures and realizing the diagram in Figure 6, left.

**Conflicts of Interest:** The authors declare no conflict of interest. The funders had no role in the design of the study; in the collection, analyses, or interpretation of data; in the writing of the manuscript; or in the decision to publish the results.

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
