# Peer review of "Integration of Passivated Gold Mirrors into Microfabricated Alkali Vapor Cells"

_coatings, doi:10.3390/coatings13101733_

Round 1

Reviewer 1 Report

The authors theoretically studied the integration of passivated gold mirrors into microfabricated alkali vapor cells. This work is interesting and has potential applications in alkali vapor cells for optically pumped magnetometers (OPMs). I would like to recommend this manuscript for publication, after the following concerns are addressed properly.

[1] The text in the left part of Figure 6 is not clear, which is hard to be seen clearly. So, I suggest the authors to change properly the font size.

[2] It is not necessary to define again the abbreviation of OPMs in the Line 26. The abbreviation has been defined in the abstract part.

[3] In Lines 46, 108, 124, 133, and137, there are two full points behind the word “found”. One of them should be deleted.

[4] I don't understand why the following sentence occurs frequently in the text:

   In Error! Reference source not found.

[5] From Line 149 to Line 160, the sentences are disorganized and cannot be understood.

Reviewer 2 Report

Manuscript review No:  coatings-2613435

 Title: Integration of passivated gold mirrors into microfabricated alkali vapor cells

 Authors: Florian Wittkämper, Theo Scholtes, Sven Linzen, Mario Ziegler, Ronny Stolz

 A. Overview

1. In this manuscript the authors propose a new technique for addition of passivated gold mirrors into vapor cells.

 2. The contents are expressed clearly; the manuscript is well organized, and it is written in reasonable English.

Though, reading of the manuscript is required as several misprints (such as Error! Reference source not found.. ) and grammar issues ca be found in the manuscript.

 3. The authors have acknowledged recent research on this topic.

 4. As long as my knowledge, the work presented is original. However, is not a remarkable new issue.

 B. Detailed analysis.

Abstract: too lengthy must be clear and objective. State briefly what you did, how did you do it, the quantitative results you and state clearly the novelty of your work.

 1. INTRODUCTION: provides an interesting approach to the subject and there are up to date references.

 C. Overall assessment

The work presented here is very interesting and has potential for further developments.

In my opinion the manuscript can be accepted for publication after minor revision.

D. Review Criteria

1. Scope of Journal

Rating: Medium

2. Novelty and Impact

Rating: LOW

3. Technical Content

Rating: Medium

4. Presentation Quality

Rating: Medium

The contents are expressed clearly; the manuscript is well organized, and it is written in reasonable English.

Though, reading of the manuscript is required as several misprints (such as Error! Reference source not found.. ) and grammar issues ca be found in the manuscript.

Reviewer 3 Report

Dear Authors,

Your article is written carelessly. It appears that you have not read the PDF file offered for review.

Please find comments to the manuscript in the attached PDF file, as sticky notes.

Kind regards.

Minor editing of English language required.

Round 2

Reviewer 1 Report

All my questions have been addressed properly, and this work can be accepted for publication.

Author Response

.

Reviewer 3 Report

Dear Authors,

Despite neither in the abstract nor in the introduction you have not explained why did you use caesium in your device, your manuscript could be published.

Kind regards.

Minor editing of English language required.
